

# Confidence intervals for the difference between the coefficients of variation of Weibull distributions for analyzing wind speed dispersion

Manussaya La-ongkaew, Sa-Aat Niwitpong and Suparat Niwitpong

Department of Applied Statistics, King Mongkut's University of Technology North Bangkok, Bangkok, Thailand

## ABSTRACT

Wind energy is an important renewable energy source for generating electricity that has the potential to replace fossil fuels. Herein, we propose confidence intervals for the difference between the coefficients of variation of Weibull distributions constructed using the concepts of the generalized confidence interval (GCI), Bayesian methods, the method of variance estimates recovery (MOVER) based on Hendricks and Robey's confidence interval, a percentile bootstrap method, and a bootstrap method with standard errors. To analyze their performances, their coverage probabilities and expected lengths were evaluated via Monte Carlo simulation. The simulation results indicate that the coverage probabilities of GCI were greater than or sometimes close to the nominal confidence level. However, when the Weibull shape parameter was small, the Bayesian- highest posterior density interval was preferable. All of the proposed confidence intervals were applied to wind speed data measured at 90-meter wind energy potential stations at various regions in Thailand.

## INTRODUCTION

Nowadays, Thailand is increasingly using alternative energy sources to oil and coal owing to the government pushing forward a policy concerning alternative energy development and efficiency. The alternative energy development in Thailand has primarily relied on energy production in domestic with emphasis on solar, wind, hydro energy, biomass, biogas, municipal Solid waste (MSW), geothermal power and biofuels including ethanol and biodiesel in term of electricity, heat and biofuels (https://www.dede.go.th/ewt_news.php?nid=47340). Wind energy is an important natural energy source for use in power generation because it has many advantages: it is clean and environmentally friendly, it reduces the level of carbon dioxide emissions that contribute toward global warming, and it offers unlimited renewable energy. In recent years, power generation from wind energy in Thailand has limitation about potential of wind speeds

Corresponding author
Sa-Aat Niwitpong, sa-aat.n@sci.kmutnb.ac.th

such as wind power density, wind speed, and the height of wind turbine (*Chancham et al., 2009*). In addition, there are problems about where to install wind turbines because wind energy is related to the topography of the installation area and wind strength variation throughout the year. Potential areas to harvest wind energy in Thailand are coastal areas and the northeastern region of Thailand. Since the coefficient of variation can be used to analyze areas with high wind speed variability, we are interested in analyzing wind datasets from the aforementioned areas in terms of the difference between coefficients of variation of Weibull distributions. The approach in this study could be useful in finding areas that are the most suitable for placing wind turbine systems to produce electricity.

In statistics, the Weibull distribution, which is named after Swedish mathematician Waloddi Weibull (*Weibull, 1951*), is a continuous probability distribution commonly used to analyze wind speed data. It has also been widely applied in engineering, industry, and weather forecasting. For example, *Chaichana et al. (2015)* studied the potential of wind energy for high-tech agricultural projects as part of the Royal Initiative at Chiang Mai province, Thailand. Moreover, *Pang, Forster & Troutt (2001)* estimated the parameters for a Weibull distribution when applied to wind speed data obtained from an observatory in Hong Kong. Previously, many researchers have estimated the parameters and functions of parameters for Weibull distributions. For instance, *Green et al. (1994)* used the Markov chain Monte Carlo (MCMC) method to estimate three parameters of a Weibull distribution. *Colosimo & Ho (1999)* derived confidence intervals for the Weibull mean lifetime based on censored reliability datasets. *Krishnamoorthy, Lin & Xia (2009)* constructed confidence intervals for the mean of Weibull distribution based on the generalized variable approach and compared this with Wald confidence intervals. In another study, *Krishnamoorthy & Lin (2010)* presented the confidence interval for stress–strength reliability involving two Weibull distributions. *Ibrahim (2010)* compared the performance of Bayesian approaches using Jeffreys' prior and an extension of Jeffreys' prior with maximum likelihood estimation to estimate the parameters of a Weibull distribution. *Kundu & Howlader (2010)* used Bayesian inference to estimate the scale parameter of an inverted Weibull distribution based on Type-II censored data. *Yalcinkaya & Birgoren (2017)* compared a confidence interval of the lower percentiles in a Weibull distribution of small samples based on maximum likelihood estimation and the Bayesian-Weibull method. *Saraiva & Suzuki (2017)* compared methods to estimate the parameters of a Weibull distribution, which are a maximum likelihood method and a Bayesian method with the Metropolis–Hastings algorithm.

The coefficient of variation is used for measuring variations in data as well as comparing the degree of variation between several datasets in which the measurement units are different. It has been used in many fields. *Faber & Korn (1991)* used the coefficient of variation to measure variation in the mean synaptic response of nerves in the central nervous system. *Banik & Kibria (2011)* used the coefficient of variation to analyze data on psychiatric disorders, while *Yosboonruang, Niwitpong & Niwitpong (2019)* used it to examine variation in rainfall data in Thailand. In this study, we focus on the coefficient of variation. Many researchers have provided confidence intervals for estimating the single and function of coefficients of variation. For example in a normal distribution: a normal coefficient
of variation (*Vangel, 1996*), common coefficient of variation (*Tian, 2005*), coefficient of variation (*Mahmoudvand & Hassani, 2009*) and (*Donner & Zou, 2010*). Moreover, *Pang et al. (2005)* proposed confidence intervals for estimating the coefficient of variation of a three-parameter Weibull distribution via a simulation-based Bayesian approach. *Niwitpong (2013)* provided confidence intervals for the coefficient of variation of a lognormal distribution with a restricted parameter space. *Buntao & Niwitpong (2013)* constructed confidence intervals for the difference between the coefficients of variation of lognormal distributions. *Sangnawakij, Niwitpong & Niwitpong (2015)* presented confidence intervals for the ratio of coefficients of variation of gamma distributions. *Yosboonruang, Niwitpong & Niwitpong (2020)* proposed confidence intervals for the difference between the coefficients of variation of delta-lognormal distributions. *La-ongkaew, Niwitpong & Niwitpong (2009)* constructed confidence interval for single coefficient of variation of Weibull distribution. It could be seen that not many researches mentioned above focused on coefficient of variation or difference between the coefficients of variation of Weibull distributions. Consequently, we extended this study to construct confidence intervals for the difference between the coefficients of variation of Weibull distributions.

Herein, we propose new confidence intervals for the difference between the coefficients of variation of Weibull distributions using the generalized confidence interval (GCI) method, Bayesian methods, the method of variance estimates recovery (MOVER) method, a percentile bootstrap method, and a bootstrap interval with standard errors, the derivations of which are given in this article. As an empirical application, we use wind speed data measured at 90-meter wind energy potential stations in Thailand. Finally, we discuss the results of our study and draw conclusions on them.

## METHODS

Let a random variable $X$ follow the 2-Parameter Weibull distribution. The probability density function of $X$ is given by

$$f(x;a,k) = \frac{k}{a}\left(\frac{x}{a}\right)^{k-1} \exp\left[-\left(\frac{x}{a}\right)^k\right], x > 0, \tag{1}$$

where the positive constant $a$ is called the scale parameter and the positive constant $k$ is called the shape parameter. The mean and the variance of $X$ are $\mathrm{E}(X) = a\Gamma\left(1+\frac{1}{k}\right)$ and $\mathrm{Var}(X) = a^2\left[\Gamma\left(1+\frac{2}{k}\right) - \left(\Gamma\left(1+\frac{1}{k}\right)\right)^2\right]$, respectively. Therefore, the coefficient of variation of $X$ can be written as

$$\mathrm{CV}(X) = \lambda = \sqrt{\frac{\Gamma\left(1+\frac{2}{k}\right)}{\left(\Gamma\left(1+\frac{1}{k}\right)\right)^2} - 1}. \tag{2}$$

Suppose that $X = (X_1, X_2, \ldots, X_n)$ be a random sample of size $n$ from Weibull distribution with scale parameter $a$ and shape parameter $k$, denoted as $Weibull(a, k)$. To estimate the parameters $a$ and $k$, the maximum likelihood estimation is applied. The maximum likelihood estimators (MLEs) can be obtained from *Cohen (1965)*, they are as follows. The

MLE $\hat{k}$ of $k$ is solution of the following equation

$$\frac{1}{\hat{k}} - \frac{\sum_{i=1}^{n}\left[x_i^{\hat{k}}\ln(x_i)\right]}{\sum(x_i^{\hat{k}})} + \frac{1}{n}\sum_{i=1}^{n}\ln(x_i) = 0 \qquad (3)$$

and the MLE $\hat{a}$ of $a$ is given by

$$\hat{a} = \left[\sum_{i=1}^{n} x_i^{\hat{k}}/n\right]^{\frac{1}{\hat{k}}}. \qquad (4)$$

For solving the Eq.(3), the Newton–Raphson iterative method can be applied. We obtained a starting value for the Newton–Raphson iterative method. Let $x_1, x_2,\ldots, x_n$ be a sample of observations from a $Weibull(a,k)$ distribution. Let $z_i = \ln(x_i), i = 1, 2, \ldots, n$. *Menon (1963)* showed that the estimator

$$\hat{k}_u = \left(\frac{\pi}{\sqrt{6}}\right)\left(\sqrt{\frac{n-1}{s^2}}\right) \qquad (5)$$

is asymptotically unbiased estimator, which is the starting value.

The following algorithm is used to compute the MLEs:

**Algorithm 1**
1. Generate $x_1, x_2,\ldots, x_n$, let $z_i = \ln(x_i)$.
2. Compute $s_1 = \sum_{i=1}^{n} z_i$, $\bar{z} = s_1/n$ and $s^2 = \sum_{i=1}^{n}(z_i - \bar{z})^2$.
3. Compute $\hat{k}_u = \left(\frac{\pi}{\sqrt{6}}\right)\left(\sqrt{\frac{n-1}{s^2}}\right)$.
4. For $l = 1$ to number of iterations.
5. Compute $w_i = x_i^{\hat{k}_u}$, $s_2 = \sum_{i=1}^{n}(w_i)$, $s_3 = \sum_{i=1}^{n}(w_i z_i)$ and $s_4 = \sum_{i=1}^{n}(w_i z_i^2)$.
6. Compute $F = \left(\frac{1}{\hat{k}_u}\right) + \bar{z} - \left(\frac{s_3}{s_2}\right)$.
7. Compute $f = \left(\frac{1}{\hat{k}_u^2}\right) + \left[(s_2 s_4) - (s_3^2)\right]/(s_2^2)$.
8. Compute $\hat{k}_u = \hat{k}_u + \left(\frac{F}{f}\right)$.
9. If ($F \leq$ error tolerance) return $\hat{k} = \hat{k}_u$.
10. End $l$ loop.
11. Compute $\hat{a}$ from Eq. (4).

Let $X_1, X_2, \ldots, X_n$ and $Y_1, Y_2, \ldots, Y_m$ be random samples from two independent Weibull distributions with scale parameters $a_X, a_Y$ and shape parameters $k_X, k_Y$, respectively. Moreover, let $\lambda_X$ be the coefficient of variation of $X$. Similarly, let $\lambda_Y$ be the coefficient of variation of $Y$. Thus, the difference between the coefficients of variation is defined by

$$\delta = \lambda_X - \lambda_Y = \sqrt{\frac{\Gamma\left(1 + \frac{2}{k_X}\right)}{\left(\Gamma\left(1 + \frac{1}{k_X}\right)\right)^2} - 1} - \sqrt{\frac{\Gamma\left(1 + \frac{2}{k_Y}\right)}{\left(\Gamma\left(1 + \frac{1}{k_Y}\right)\right)^2} - 1}. \qquad (6)$$

And the estimator of $\delta$ is given by

$$\hat{\delta} = \hat{\lambda}_X - \hat{\lambda}_Y = \sqrt{\frac{\Gamma\left(1 + \frac{2}{\hat{k}_X}\right)}{\left(\Gamma\left(1 + \frac{1}{\hat{k}_X}\right)\right)^2} - 1} - \sqrt{\frac{\Gamma\left(1 + \frac{2}{\hat{k}_Y}\right)}{\left(\Gamma\left(1 + \frac{1}{\hat{k}_Y}\right)\right)^2} - 1}. \qquad (7)$$

## THE GENERALIZED CONFIDENCE INTERVAL

The generalized confidence interval (GCI) was introduced by *Weerahandi (1993)*. The most important concept of GCI is the generalized pivotal quantity (GPQ) defined as

**Definition** Suppose that $X = (X_1, X_2, \ldots, X_n)$ be a random variables from distribution, which depends on a parameter of interest $\varphi$, and a nuisance parameter $\gamma$. Furthermore, suppose that $x = (x_1, x_2, \ldots, x_n)$ be the observed value of X. To obtain a GCI for $\varphi$, a GPQ $R(X; x, \varphi, \gamma)$ is also required to satisfy the following two conditions.

(i) For a fixed x, the distribution of $R(X; x, \varphi, \gamma)$ is free of unknown parameters.

(ii) The observed value of $R(X; x, \varphi, \gamma)$ at $X = x$, denoted as $r(X; x, \varphi, \gamma)$ does not depend on nuisance parameters.

Let $R_\varphi(\alpha/2)$ and $R_\varphi(1 - \alpha/2)$ are the $100(\alpha/2)$-th and $100(1 - \alpha/2)$-th percentile of $R_\varphi(X; x)$, respectively. The $100(1 - \alpha)\%$ two-sided confidence interval based on GCI for parameter of interest is given by $[R_\varphi(\alpha/2), R_\varphi(1 - \alpha/2)]$. Note that $1 - \alpha$ is the probability that population parameter will be in the interval. The $100(1 - \alpha)\%$ confidence interval will include the true value of the population parameter with probability $1 - \alpha$, i.e., if $\alpha = 0.05$, that probability is about 0.95 that the 95% confidence interval will include the true population parameter.

For Weibull distribution, *Thoman, Bain & Antle (1969)* showed that the distributions of $\frac{\hat{k}}{k}$ and $\hat{k} \ln\left(\frac{\hat{a}}{a}\right)$ do not depend on $a$ and $k$, where $\hat{a}$ and $\hat{k}$ are the MLEs of $a$ and $k$, respectively. Let $\hat{a^*}$ and $\hat{k^*}$ be the MLEs based on a sample of size $n$ from a *Weibull*$(1, 1)$. Then $\frac{\hat{k}}{k} \sim \hat{k^*}$ and $\hat{k} \ln\left(\frac{\hat{a}}{a}\right) \sim \hat{k^*} \ln(\hat{a^*})$. Both distributions do not depend on any parameters. Hence, they are pivotal quantity.

*Krishnamoorthy, Mukherjee & Guo (2007)* presented the GPQs of scale and shape parameters from Weibull distribution. Let $\hat{a}_0$ and $\hat{k}_0$ be the observed values of $\hat{a}$ and $\hat{k}$. The GPQs of the parameters are given by

$$R_k = \frac{k}{\hat{k}} \hat{k}_0 = \frac{\hat{k}_0}{\hat{k^*}} \tag{8}$$

and

$$R_a = \left(\frac{a}{\hat{a}}\right)^{\frac{\hat{k}}{\hat{k}_0}} \hat{a}_0 = \left(\frac{1}{\hat{a^*}}\right)^{\frac{\hat{k^*}}{\hat{k}_0}} \hat{a}_0. \tag{9}$$

From the random variables $X$ and $Y$, the GPQs for $k_X$ and $k_Y$ are given by

$$R_{k_X} = \frac{\hat{k}_{X_0}}{\hat{k^*_X}} \tag{10}$$

and

$$R_{k_Y} = \frac{\hat{k}_{Y_0}}{\hat{k^*_Y}}. \tag{11}$$

Thus, the GPQ for $\delta$ is defined by

$$R_\delta = R_{\lambda_X} - R_{\lambda_Y} = \sqrt{\frac{\Gamma\left(1+\frac{2}{R_{k_X}}\right)}{\left(\Gamma\left(1+\frac{1}{R_{k_X}}\right)\right)^2} - 1} - \sqrt{\frac{\Gamma\left(1+\frac{2}{R_{k_Y}}\right)}{\left(\Gamma\left(1+\frac{1}{R_{k_Y}}\right)\right)^2} - 1}. \qquad (12)$$

Therefore, the $100\,(1-\alpha)\%$ two-sided confidence interval for the difference between the coefficients of variation of Weibull distributions based on GCI is

$$CI_{\delta(gci)} = \left[L_{\delta(gci)}, U_{\delta(gci)}\right] = \left[R_\delta(\alpha/2), R_\delta(1-\alpha/2)\right], \qquad (13)$$

where $R_\delta(\alpha/2)$ denotes the $100(\alpha/2)$-th percentile of $R_\delta$.

The following algorithm is used to construct confidence interval based on GCI for the difference between the coefficients of variation of Weibull distributions:

**Algorithm 2**
1. Compute $\hat{a}_X$, $\hat{a}_Y$, $\hat{k}_X$ and $\hat{k}_Y$ from Algorithm 1.
2. Generate $X_1^*, X_2^*, \ldots, X_n^*$ and $Y_1^*, Y_2^*, \ldots, Y_m^*$ from $Weibull(1,1)$.
3. Compute $\hat{a}_X^*$, $\hat{a}_Y^*$, $\hat{k}_X^*$ and $\hat{k}_Y^*$ from Algorithm 1.
4. Compute $R_{k_X}$ and $R_{k_Y}$ from Eqs. (10) and (11).
5. Compute $R_\delta$ from Eq. (12).
6. Repeat steps 1-5 for $q$ times, where $q$ is the number of generalized computation.
7. Compute 95% confidence interval based on GCI, as given in Eq. (13).

## THE BOOTSTRAP CONFIDENCE INTERVALS

The bootstrap method was introduced by *Efron & Tibshirani (1993)*. This is a resampling method to determine precision measures for statistical estimation.

Let $\mathbf{x}_i = (x_1, x_2, \ldots, x_n)$ and $\mathbf{y}_i = (y_1, y_2, \ldots, y_m)$ be random samples of size $n$ and $m$ from Weibull distributions, and let $\mathbf{x}_i^{*b} = \left(x_1^{*b}, x_2^{*b}, \ldots, x_n^{*b}\right)$ and $\mathbf{y}_i^{*b} = \left(y_1^{*b}, y_2^{*b}, \ldots, y_m^{*b}\right)$ be bootstrapped samples drawn with replacement from the original data, using the same sample sizes. After resampling $B$ bootstrap samples, the difference between the coefficients of variation are calculated in each bootstrap sample, as follows: $\delta^{*b} = \lambda_X^{*b} - \lambda_Y^{*b}$, $b = 1, 2, \ldots, B$.

### The percentile bootstrap confidence interval

The percentile bootstrap confidence interval is based on the percentile of the distribution of the bootstrapped replications. The value of the bootstrap statistic, $\delta^{*b}$ are ordered from the smallest to the largest.

Therefore, the $100\,(1-\alpha)\%$ two-sided confidence interval for the difference between the coefficients of variation of Weibull distributions based on percentile bootstrap is given by

$$CI_{\delta(pb)} = \left[L_{\delta(pb)}, U_{\delta(pb)}\right] = \left[\delta^{*b}(\alpha/2), \delta^{*b}(1-\alpha/2)\right], \qquad (14)$$

where $\delta^{*b}(\alpha/2)$ denotes the $100(\alpha/2)$-th percentile of $\delta^{*b}$.

The following algorithm is used to construct confidence interval based on percentile bootstrap for the difference between the coefficients of variation of Weibull distributions:

**Algorithm 3**
1. Generate $X_1, X_2, \ldots, X_n$ from $Weibull(a_X, k_X)$ and $Y_1, Y_2, \ldots, Y_m$ from $Weibull(a_Y, k_Y)$.
2. For $b = 1$.
3. Resampling bootstrap samples $X_1^*, X_2^*, \ldots, X_n^*$ from $X_1, X_2, \ldots, X_n$ and compute $\lambda_X^{*b}$.
4. Resampling bootstrap samples $Y_1^*, Y_2^*, \ldots, Y_m^*$ from $Y_1, Y_2, \ldots, Y_m$ and compute $\lambda_Y^{*b}$.
5. Compute $\delta^{*b} = \lambda_X^{*b} - \lambda_Y^{*b}$.
6. Repeat steps 3-5 for $B$ times, where $B$ is the number of bootstrap sample.
7. Sort $\delta^{*b}$ from the smallest to the largest.
8. Compute 95% confidence interval based on percentile bootstrap, as given in Eq. (14).

**The bootstrap confidence interval with standard errors**
From the $B$ bootstrap statistic denoted as $\delta^{*b}, b = 1, 2, \ldots, B$, we can calculate the standard error of a statistic. They can be estimated using the standard deviation of the bootstrap distribution.

Therefore, the $100(1 - \alpha)\%$ two-sided confidence interval for the difference between the coefficients of variation of Weibull distributions based on bootstrap confidence interval with standard errors can be written as

$$CI_{\delta(bs)} = \left[L_{\delta(bs)}, U_{\delta(bs)}\right] = \left[\hat{\delta} - Z_{(\alpha/2)}SE, \hat{\delta} + Z_{(\alpha/2)}SE\right], \tag{15}$$

where $SE$ is the standard error of a statistic.

The following algorithm is used to construct confidence interval based on bootstrap confidence interval with standard errors for the difference between the coefficients of variation of Weibull distributions:

**Algorithm 4**
1. Generate $X_1, X_2, \ldots, X_n$ from $Weibull(a_X, k_X)$ and $Y_1, Y_2, \ldots, Y_m$ from $Weibull(a_Y, k_Y)$.
2. For $b = 1$.
3. Resampling bootstrap samples $X_1^*, X_2^*, \ldots, X_n^*$ from $X_1, X_2, \ldots, X_n$ and compute $\lambda_X^{*b}$.
4. Resampling bootstrap samples $Y_1^*, Y_2^*, \ldots, Y_m^*$ from $Y_1, Y_2, \ldots, Y_m$ and compute $\lambda_Y^{*b}$.
5. Compute $\delta^{*b} = \lambda_X^{*b} - \lambda_Y^{*b}$.
6. Repeat steps 3-5 for $B$ times, where $B$ is the number of bootstrap sample.
7. Compute $SE$.
8. Compute 95% confidence interval based on bootstrap confidence interval with standard errors, as given in Eq. (15).

## THE METHOD OF VARIANCE ESTIMATES RECOVERY

According to *Donner & Zou (2010)*, this approach can be used to construct a confidence interval for a function of two parameters, $\lambda_X - \lambda_Y$, defined as

$$CI_m = [L_m, U_m], \tag{16}$$

where the lower limit and upper limit for $\hat{\lambda}_X - \hat{\lambda}_Y$ are given by

$$L_m = \left(\hat{\lambda}_X - \hat{\lambda}_Y\right) - \sqrt{(\hat{\lambda}_X - l_X)^2 + (u_Y - \hat{\lambda}_Y)^2} \tag{17}$$

$$U_m = (\hat{\lambda}_X - \hat{\lambda}_Y) + \sqrt{(u_X - \hat{\lambda}_X)^2 + (\hat{\lambda}_Y - l_Y)^2}. \tag{18}$$

*Hendricks & Robey (1936)* presented confidence intervals for $\lambda_X$ and $\lambda_Y$ defined as

$$(l_{X.HR}, u_{X.HR}) = \left( \hat{\lambda}_X - t_{(\alpha/2,n-1)} \frac{\hat{\lambda}_X}{\sqrt{2n}}, \hat{\lambda}_X + t_{(\alpha/2,n-1)} \frac{\hat{\lambda}_X}{\sqrt{2n}} \right) \tag{19}$$

and

$$(l_{Y.HR}, u_{Y.HR}) = \left( \hat{\lambda}_Y - t_{(\alpha/2,m-1)} \frac{\hat{\lambda}_Y}{\sqrt{2m}}, \hat{\lambda}_Y + t_{(\alpha/2,m-1)} \frac{\hat{\lambda}_Y}{\sqrt{2m}} \right), \tag{20}$$

where $t_{(\alpha/2,n-1)}$ and $t_{(\alpha/2,m-1)}$ denote the $100(\alpha/2)$-th percentile of t-distribution with $n-1$ and $m-1$ degrees of freedom, respectively. To construct a confidence interval for the difference between the coefficients of variation of Weibull distributions based on MOVER with Hendricks and Robey's confidence interval, we substitute $l_{X.HR}$, $u_{X.HR}$, $l_{Y.HR}$ and $u_{Y.HR}$ for $l_X$, $u_X$, $l_Y$ and $u_Y$, respectively.

Therefore, the $100(1-\alpha)\%$ two-sided confidence interval for the difference between the coefficients of variation of Weibull distributions based on MOVER with Hendricks and Robey's confidence interval becomes

$$CI_{\delta(m.HR)} = \left[ L_{\delta(m.HR)}, U_{\delta(m.HR)} \right], \tag{21}$$

where

$$L_{\delta(m.HR)} = (\hat{\lambda}_X - \hat{\lambda}_Y) - \sqrt{(\hat{\lambda}_X - l_{X.HR})^2 + (u_{Y.HR} - \hat{\lambda}_Y)^2} \tag{22}$$

$$U_{\delta(m.HR)} = (\hat{\lambda}_X - \hat{\lambda}_Y) + \sqrt{(u_{X.HR} - \hat{\lambda}_X)^2 + (\hat{\lambda}_Y - l_{Y.HR})^2}. \tag{23}$$

The following algorithm is used to construct confidence interval based on MOVER for the difference between the coefficients of variation of Weibull distributions:

**Algorithm 5**
1. Generate $X_1, X_2, \ldots, X_n$ from *Weibull*$(a_X, k_X)$ and $Y_1, Y_2, \ldots, Y_m$ from *Weibull*$(a_Y, k_Y)$.
2. Compute the intervals for $\lambda_X$ and $\lambda_Y$ from Eqs. (19) and (20).
3. Compute 95% confidence interval based on MOVER, as given in Eq. (21).

## THE BAYESIAN CONFIDENCE INTERVALS

Here, we derive Bayesian estimates of the parameters of a Weibull distribution. Once again, for random variable $X = (X_1, X_2, \ldots, X_n)$ from a Weibull distribution, we transform scale parameter $a' = (\frac{1}{a})^k$, and so we can provide its probability density function in another form as follows:

$$f(x; a', k) = a' k x^{k-1} \exp\left[-a' x^k\right], x > 0. \tag{24}$$

It is assumed that the prior distributions for the scale and shape parameters are gamma with hyperparameters $(v_1, v_2, z_1, z_2)$:

$$\pi(a') \sim gamma(v_1, z_1) \tag{25}$$

and

$$\pi(k) \sim gamma(v_2, z_2). \tag{26}$$

Hence, using Bayes' theorem, the joint posterior density function of $a'$ and $k$ is given by

$$\pi(a', k|x) \propto L(a', k|x)\pi(a')\pi(k), \tag{27}$$

where $L(a', k|x)$ is a likelihood function.

However, the posterior distributions cannot be computed, and so we used the MCMC method with Gibbs sampling to provide them for the parameters. We used the Gibbs sampling procedure to generate samples from the joint posterior distribution, as given in Eq. (27). The conditional posterior distributions of parameters are

$$\pi(a'|k, x) \sim gamma\left(v_2 + n, z_2 + \sum x^k\right) \tag{28}$$

and

$$\pi(k|a', x) \propto k^{v_1 + n - 1} \exp\left[-v_1 k - a' \sum x^k\right]. \tag{29}$$

For Eq. (29), we used the Metropolis-Hasting (MH) algorithm to update shape parameter $k$. Moreover, the random walk Metropolis (RWM) and Gibbs sampling methods are applied as follows (*Geman & Geman, 1984*):

**Algorithm 6** The Gibbs algorithm
1. Take the initial value of parameter $(a'^{(0)}, k^{(0)})$.
2. Generate $a'^{(t)} \sim gamma(v_2 + n, z_2 + \sum x^{k^{(t-1)}})$.
3. Using Random walk Metropolis (RWM) algorithm for calculate $k^{(t)}$.
4. Repeat step 2–3 for $T$ times, where $T$ is the number of replications of MCMC.
5. Burn in 1000 samples and compute the parameter of interest.

**Algorithm 7** The random walk Metropolis (RWM)
1. Start with $(a'^{(t)}, k^{(t-1)})$.
2. Generate $\varepsilon \sim N\left(0, \sigma_k^2\right)$.
3. Compute $k^* = k^{(t-1)} + \varepsilon$.
4. Compute $A_k = \frac{L(k^*, a'|x)\pi(k^*)}{L(k, a'|x)\pi(k)}$.
5. Generate $u \sim U(0, 1)$.
6. If $u \leq min(1, A_k)$ set $k^{(t)} = k^*$ and If $u > min(1, A_k)$ set $k^{(t)} = k^{(t-1)}$.

## The Bayesian-MCMC

Once again, let $X$ and $Y$ be random variables from Weibull distributions. First, we used Algorithms 6 and 7 to calculate the difference between the coefficients of variation denoted as $\delta^t, t = 1, 2, \ldots, T$, based on the Bayesian-MCMC.

Therefore, the $100(1 - \alpha)\%$ two-sided confidence interval for the difference between the coefficients of variation of Weibull distributions based on the Bayesian-MCMC is given by

$$CI_{\delta(MCMC)} = \left[L_{\delta(MCMC)}, U_{\delta(MCMC)}\right] = \left[\delta^t(\alpha/2), \delta^t(1 - \alpha/2)\right], \tag{30}$$

where $\delta^t(\alpha/2)$ denotes the $100(\alpha/2)$-th percentile of $\delta^t$.

The following algorithm is used to construct confidence interval based on Bayesian-MCMC for the difference between the coefficients of variation of Weibull distributions:

**Algorithm 8**
1. Calculate the initial value of parameter $(a'^{(0)}, k^{(0)})$.
2. Calculate $\delta^t$ from Algorithms 6 and 7.
3. Compute 95% confidence interval based on Bayesian-MCMC, as given in Eq. (30).

### The Bayesian-Highest Posterior Density (HPD) Interval

Here, the assumption is that the density of every point inside the interval is greater than that of every point outside the interval, and it is also the shortest interval (*Box & Tiao, 2011*). In this article, the HPD interval was computed by using the R package with HDInterval.

Therefore, the $100(1-\alpha)$% two-sided confidence interval for the difference between the coefficients of variation of Weibull distributions based on the Bayesian-HPD interval can be written as

$$CI_{\delta(HPD)} = \left[ L_{\delta(HPD)}, U_{\delta(HPD)} \right]. \tag{31}$$

The following algorithm is used to construct confidence interval based on Bayesian-HPD for the difference between the coefficients of variation of Weibull distributions:

**Algorithm 9**
1. Calculate the initial value of parameter $(a'^{(0)}, k^{(0)})$.
2. Calculate $\delta^t$ from Algorithms 6 and 7.
3. Compute 95% confidence interval based on Bayesian-HPD interval, as given in Eq. (31).

## RESULTS

A simulation study was conducted to compare the performances of the six confidence intervals for the difference between the coefficients of variation of Weibull distributions: GCI, the Bayesian-MCMC method, the Bayesian-HPD interval, MOVER based on Hendricks and Robey's confidence interval, the percentile bootstrap method, and the bootstrap confidence interval with standard errors. The R program was used to estimate the coverage probabilities and expected lengths of the proposed confidence intervals using the following parameter settings:

Scale parameter $a_X = a_Y = 0.5$ and 2.
Shape parameter $k_X = 1$ and $k_Y = 0.5, 1, 2, 2.5, 4$ and 9, with the differences between the coefficients of variation of $-1.2360, 0, 0.4772, 0.5720, 0.7194$ and $0.8671$, respectively.
Sample sizes $(n, m) = (10,10), (10,20), (20,20), (30,30), (30,50), (50,50), (50,100)$ and $(100,100)$.
Hyperparameters $v_1 = v_2 = z_1 = z_2 = 0.1$.

The number of replications for each situation was 5,000, along with 2,500 pivotal quantities for GCI. Moreover, we used 500 bootstrap samples for the bootstrap methods and we generated 20,000 realizations of MCMC using the Gibbs and RWM algorithms with a burn-in of 1000. The following performance indicators were used to determine

the best-performing method: a coverage probability of greater than or equal to 0.95 (the nominal confidence level) and the shortest expected length.

The following algorithm is used to estimate the coverage probability and expected length:

**Algorithm 10**

1. Set $M, q, B, T, n, m, a_X, k_X, a_Y$ and $k_Y$.
2. Generate $X_1, X_2, \ldots, X_n$ from $Weibull(a_X, k_X)$ and $Y_1, Y_2, \ldots, Y_m$ from $Weibull(a_Y, k_Y)$, respectively.
3. Use Algorithm 2 to construct generalized confidence interval ($CI_{\delta(gci)}$).
4. Use Algorithm 3 to construct percentile bootstrap confidence interval ($CI_{\delta(pb)}$).
5. Use Algorithm 4 to construct bootstrap confidence interval with standard errors ($CI_{\delta(bs)}$).
6. Use Algorithm 5 to construct MOVER based on Hendricks and Robey's confidence interval ($CI_{\delta (m.HR)}$).
7. Use Algorithm 8 to construct Baysian-MCMC confidence interval ($CI_{\delta(MCMC)}$).
8. Use Algorithm 9 to construct Baysian-HPD confidence interval ($CI_{\delta(HPD)}$).
9. If ($L \leq \delta \leq U$), then set $P = 1$, else set $P = 0$.
10. Compute ($U - L$).
11. Repeat steps 2-10 for $M$ times.
12. Compute mean of $P$ for the coverage probability.
13. Compute mean of ($U - L$) for the expected length.

For $a = 0.5$ (Table 1 and Fig. 1), the simulation results indicate that the coverage probabilities of $CI_{\delta(gci)}$ were greater than or sometimes close to the nominal confidence level of 0.95 in almost all cases. $CI_{\delta(pb)}$ yielded coverage probabilities greater than 0.95 when $k_Y = 1$ for $(n, m) = (10, 10)$ and $(20, 20)$. For the Bayesian methods, the coverage probabilities of $CI_{\delta(MCMC)}$ and $CI_{\delta(HPD)}$ were greater than or sometimes close to the nominal confidence level, while the expected lengths of $CI_{\delta(HPD)}$ for $(n, m) = (10, 10)$ and $k_Y = 0.5, 2, 4$; $(n, m) = (20, 20)$ and $k_Y = 0.5$; $(n, m) = (10, 20), (30, 30), (30, 50), (50, 50), (50, 100)$ and $k_Y = 0.5, 1$; and $(n, m) = (100, 100)$ and $k_Y = 1$ were the shortest. Moreover, $CI_{\delta(bs)}$ and $CI_{\delta(m.HR)}$ yielded coverage probabilities under 0.95 for all cases.

From the simulation results for $a = 2$ (Table 2 and Fig. 2), $CI_{\delta(gci)}$ performed well in terms of coverage probability in almost all cases, while those of $CI_{\delta(pb)}, CI_{\delta(bs)}$ and $CI_{\delta(m.HR)}$ were the same as for $a = 0.5$. $CI_{\delta(pb)}$ yielded coverage probabilities of under 0.95 in almost all cases except when $(n, m) = (10, 10), (20, 20)$ and $k_Y = 0.5$. For the Bayesian methods, the coverage probabilities of $CI_{\delta(HPD)}$ were greater than 0.95 and its expected lengths were shortest for $(n, m) = (10, 20), (30, 50), (50, 100)$ and $k_Y = 0.5, 1$; $(n, m) = (20, 20)$ and $k_Y = 2$; $(n, m) = (30, 30)$ and $k_Y = 1, 2, 2.5$; $(n, m) = (50, 50)$ and $k_Y = 0.5, 1, 2.5$; and $(n, m) = (100, 100)$ for all $k_Y$ except for $k_Y = 4$ and $9$.

**Table 1 Comparison results of the 95% two-sided confidence intervals for the difference between the coefficients of variation of Weibull distributions for $a = 0.5$.**

| $(n, m)$ | $k$ | Coverage probability (Expected length) | | | | | |
|---|---|---|---|---|---|---|---|
| | | $CI_{\delta(gci)}$ | $CI_{\delta(pb)}$ | $CI_{\delta(bs)}$ | $CI_{\delta(m.HR)}$ | $CI_{\delta(MCMC)}$ | $CI_{\delta(HPD)}$ |
| (10,10) | 0.5 | 0.9568 | 0.6324 | 0.6956 | 0.8430 | 0.9392 | 0.9514 |
| | | (8.1452) | (1.6874) | (1.7114) | (2.3678) | (5.4003) | (4.6368) |
| | 1 | 0.9494 | 0.9614 | 0.9032 | 0.9610 | 0.9388 | 0.9706 |
| | | (2.3498) | (1.1568) | (1.1587) | (1.3535) | (1.8406) | (1.7986) |
| | 2 | 0.9510 | 0.9046 | 0.8860 | 0.9410 | 0.9454 | 0.9578 |
| | | (1.7089) | (0.9219) | (0.9248) | (1.0787) | (1.3696) | (1.3018) |
| | 2.5 | 0.9492 | 0.8710 | 0.8610 | 0.9152 | 0.9370 | 0.9400 |
| | | (1.6386) | (0.8863) | (0.8894) | (1.0359) | (1.3144) | (1.2340) |
| | 4 | 0.9580 | 0.8604 | 0.8526 | 0.9026 | 0.9360 | 0.9510 |
| | | (1.5526) | (0.8366) | (0.8404) | (0.9827) | (1.2451) | (1.1469) |
| | 9 | 0.9510 | 0.8374 | 0.8334 | 0.8854 | 0.9120 | 0.8764 |
| | | (1.5173) | (0.8076) | (0.8105) | (0.9515) | (1.2218) | (1.1060) |
| (10,20) | 0.5 | 0.9564 | 0.7056 | 0.7924 | 0.8598 | 0.9476 | 0.9664 |
| | | (3.6785) | (1.6034) | (1.6216) | (1.7373) | (3.1102) | ( 2.9756) |
| | 1 | 0.9504 | 0.9378 | 0.8948 | 0.9344 | 0.9416 | 0.9616 |
| | | (1.8251) | (1.0235) | (1.0258) | (1.1510) | (1.5018) | (1.4525) |
| | 2 | 0.9536 | 0.8620 | 0.8536 | 0.9018 | 0.9406 | 0.9362 |
| | | (1.5737) | (0.8642) | (0.8664) | (1.0037) | (1.2677) | (1.1743) |
| | 2.5 | 0.9543 | 0.8480 | 0.8448 | 0.8952 | 0.9382 | 0.9300 |
| | | (1.5418) | (0.8407) | (0.8432) | (0.9810) | (1.2400) | (1.1376) |
| | 4 | 0.9488 | 0.8380 | 0.8312 | 0.8836 | 0.9386 | 0.9154 |
| | | (1.5419) | (0.8264) | (0.8302) | (0.9668) | (1.2290) | (1.1124) |
| | 9 | 0.9482 | 0.8262 | 0.8250 | 0.8786 | 0.9204 | 0.8794 |
| | | (1.5300) | (0.8106) | (0.8144) | (0.9544) | (1.2208) | (1.0958) |
| (20,20) | 0.5 | 0.9524 | 0.6292 | 0.7778 | 0.8282 | 0.9480 | 0.9536 |
| | | (3.2987) | (1.4974) | (1.5220) | (1.5855) | (2.8839) | (2.6795) |
| | 1 | 0.9530 | 0.9564 | 0.9170 | 0.9450 | 0.9444 | 0.9656 |
| | | (1.2350) | (0.8758) | (0.8814) | (0.9149) | (1.1264) | (1.1166) |
| | 2 | 0.9522 | 0.9024 | 0.9010 | 0.9330 | 0.9434 | 0.9486 |
| | | (0.9234) | (0.6836) | (0.6889) | (0.7236) | (0.8471) | (0.8260) |
| | 2.5 | 0.9526 | 0.8900 | 0.8918 | 0.9276 | 0.9458 | 0.9470 |
| | | (0.8915) | (0.6566) | (0.6615) | (0.6969) | (0.8178) | (0.7917) |
| | 4 | 0.9534 | 0.8730 | 0.8870 | 0.9166 | 0.9430 | 0.9344 |
| | | (0.8589) | (0.6348) | (0.6403) | (0.6666) | (0.7866) | (0.7532) |
| | 9 | 0.9522 | 0.8652 | 0.8720 | 0.9114 | 0.9194 | 0.8866 |
| | | (0.8395) | (0.6149) | (0.6225) | (0.6479) | (0.7729) | (0.7340) |
| (30,30) | 0.5 | 0.9506 | 0.6428 | 0.8124 | 0.8228 | 0.9448 | 0.9501 |
| | | (2.3298) | (1.3805) | (1.4109) | (1.2717) | (2.1510) | (2.0452) |
| | 1 | 0.9522 | 0.9426 | 0.9244 | 0.9382 | 0.9480 | 0.9618 |
| | | (0.9229) | (0.7415) | (0.7481) | (0.7346) | (0.8728) | (0.8678) |

**Table 1** (*continued*)

| $(n, m)$ | $k$ | Coverage probability (Expected length) | | | | | |
|---|---|---|---|---|---|---|---|
| | | $CI_{\delta(gci)}$ | $CI_{\delta(pb)}$ | $CI_{\delta(bs)}$ | $CI_{\delta(m.HR)}$ | $CI_{\delta(MCMC)}$ | $CI_{\delta(HPD)}$ |
| | 2 | 0.9510 | 0.8972 | 0.9082 | 0.9298 | 0.9462 | 0.9482 |
| | | (0.6990) | (0.5770) | (0.5819) | (0.5820) | (0.6621) | (0.6506) |
| | 2.5 | 0.9482 | 0.8918 | 0.9064 | 0.9280 | 0.9452 | 0.9476 |
| | | (0.6787) | (0.5581) | (0.5629) | (0.5625) | (0.6430) | (0.6287) |
| | 4 | 0.9532 | 0.8776 | 0.9036 | 0.9212 | 0.9476 | 0.9394 |
| | | (0.6526) | (0.5364) | (0.5429) | (0.5370) | (0.6171) | (0.5991) |
| | 9 | 0.9446 | 0.8602 | 0.8814 | 0.9036 | 0.9080 | 0.8784 |
| | | (0.6365) | (0.5187) | (0.5246) | (0.5207) | (0.6040) | (0.5835) |
| (30,50) | 0.5 | 0.9574 | 0.6856 | 0.8760 | 0.8416 | 0.9530 | 0.9634 |
| | | (1.7122) | (1.2865) | (1.3262) | (1.0347) | (1.6309) | (1.5927) |
| | 1 | 0.9400 | 0.9356 | 0.9210 | 0.9240 | 0.9360 | 0.9500 |
| | | (0.8011) | (0.6768) | (0.6822) | (0.6526) | (0.7652) | (0.7591) |
| | 2 | 0.9494 | 0.8848 | 0.8996 | 0.9226 | 0.9452 | 0.9448 |
| | | (0.6690) | (0.5518) | (0.5568) | (0.5556) | (0.6354) | (0.6203) |
| | 2.5 | 0.9462 | 0.8812 | 0.8948 | 0.9160 | 0.9452 | 0.9418 |
| | | (0.6581) | (0.5399) | (0.5455) | (0.5440) | (0.6246) | (0.6074) |
| | 4 | 0.9450 | 0.8696 | 0.8880 | 0.9098 | 0.9414 | 0.9382 |
| | | (0.6447) | (0.5297) | (0.5359) | (0.5289) | (0.6095) | (0.5897) |
| | 9 | 0.9498 | 0.8608 | 0.8878 | 0.9056 | 0.9214 | 0.8936 |
| | | (0.6306) | (0.5144) | (0.5208) | (0.5170) | (0.5981) | (0.5769) |
| (50,50) | 0.5 | 0.9520 | 0.6502 | 0.8596 | 0.8202 | 0.9474 | 0.9504 |
| | | (1.6289) | (1.2294) | (1.2703) | (0.9739) | (1.5586) | (1.5103) |
| | 1 | 0.9474 | 0.9368 | 0.9330 | 0.9278 | 0.9456 | 0.9552 |
| | | (0.6699) | (0.6106) | (0.6165) | (0.5626) | (0.6485) | (0.6460) |
| | 2 | 0.9456 | 0.9058 | 0.9208 | 0.9304 | 0.9460 | 0.9480 |
| | | (0.5171) | (0.4724) | (0.4766) | (0.4489) | (0.5010) | (0.4951) |
| | 2.5 | 0.9512 | 0.8900 | 0.9178 | 0.9264 | 0.9448 | 0.9450 |
| | | (0.4992) | (0.4545) | (0.4586) | (0.4319) | (0.4840) | (0.4770) |
| | 4 | 0.9558 | 0.8844 | 0.9154 | 0.9228 | 0.9482 | 0.9412 |
| | | (0.4796) | (0.4384) | (0.4432) | (0.4119) | (0.4640) | (0.4552) |
| | 9 | 0.9458 | 0.8744 | 0.9082 | 0.9148 | 0.9132 | 0.8908 |
| | | (0.4695) | (0.4246) | (0.4299) | (0.4008) | (0.4564) | (0.4463) |
| (50,100) | 0.5 | 0.9532 | 0.7166 | 0.9172 | 0.8386 | 0.9518 | 0.9580 |
| | | (1.1384) | (1.1032) | (1.1514) | (0.7422) | (1.1119) | (1.0987) |
| | 1 | 0.9584 | 0.9364 | 0.9448 | 0.9346 | 0.9538 | 0.9610 |
| | | (0.5672) | (0.5404) | (0.5448) | (0.4843) | (0.5522) | (0.5486) |
| | 2 | 0.9480 | 0.8898 | 0.9176 | 0.9244 | 0.9460 | 0.9484 |
| | | (0.4897) | (0.4464) | (0.4505) | (0.4230) | (0.4759) | (0.4684) |
| | 2.5 | 0.9518 | 0.8820 | 0.9098 | 0.9116 | 0.9496 | 0.9442 |
| | | (0.4814) | (0.4380) | (0.4425) | (0.4145) | (0.4668) | (0.4581) |

**Table 1** (*continued*)

| (*n*, *m*) | *k* | Coverage probability (Expected length) | | | | | |
|---|---|---|---|---|---|---|---|
| | | $CI_{\delta(gci)}$ | $CI_{\delta(pb)}$ | $CI_{\delta(bs)}$ | $CI_{\delta(m.HR)}$ | $CI_{\delta(MCMC)}$ | $CI_{\delta(HPD)}$ |
| | 4 | 0.9494 | 0.8734 | 0.9082 | 0.9164 | 0.9466 | 0.9418 |
| | | (0.4700) | (0.4309) | (0.4361) | (0.4041) | (0.4567) | (0.4471) |
| | 9 | 0.9486 | 0.8704 | 0.9006 | 0.9074 | – | – |
| | | (0.4648) | (0.4206) | (0.4253) | (0.3974) | – | – |
| (100,100) | 0.5 | 0.9488 | 0.6768 | 0.9102 | 0.8198 | 0.9468 | 0.9472 |
| | | (1.0705) | (1.0617) | (1.1132) | (0.6829) | (1.0486) | (1.0308) |
| | 1 | 0.9496 | 0.9362 | 0.9548 | 0.9266 | 0.9496 | 0.9524 |
| | | (0.4523) | (0.4660) | (0.4692) | (0.3949) | (0.4453) | (0.4440) |
| | 2 | 0.9434 | 0.9076 | 0.9346 | 0.9256 | 0.9444 | 0.9460 |
| | | (0.3514) | (0.3558) | (0.3584) | (0.3152) | (0.3461) | (0.3437) |
| | 2.5 | 0.9514 | 0.9060 | 0.9386 | 0.9286 | 0.9486 | 0.9492 |
| | | (0.3394) | (0.3435) | (0.3462) | (0.3032) | (0.3341) | (0.3313) |
| | 4 | 0.9532 | 0.9004 | 0.9390 | 0.9270 | 0.9516 | 0.9484 |
| | | (0.3269) | (0.3344) | (0.3376) | (0.2896) | (0.3214) | (0.3179) |
| | 9 | 0.9514 | 0.8954 | 0.9336 | 0.9224 | – | – |
| | | (0.3188) | (0.3223) | (0.3250) | (0.2809) | – | – |

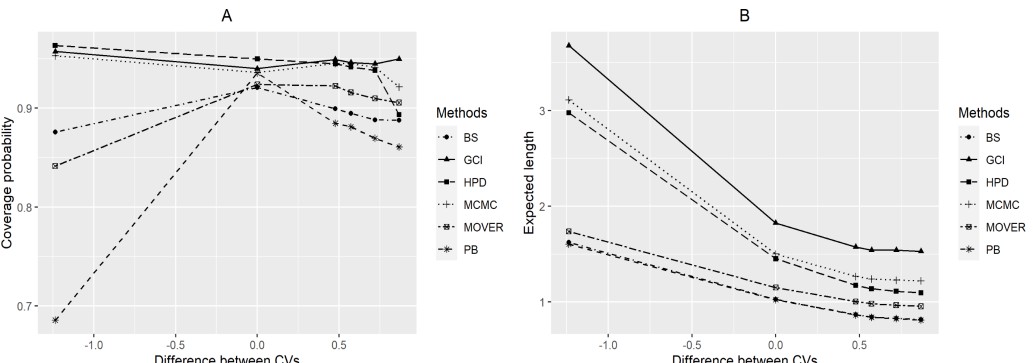

**Figure 1** Graphs to compare the performance of the methods with varying difference between CVs in terms of (A) coverage probability and (B) expected length.

# EMPIRICAL APPLICATION OF THE DERIVED CONFIDENCE INTERVALS

## Example 1

Wind speed data set were collected at 90-meter wind energy potential stations in Trad and Chonburi provinces in 2016 by the Department of Alternative Energy Development and Efficiency, Ministry of Energy (https://www.dede.go.th/ewt_news.php?nid=47706). These are coastal provinces in eastern Thailand that have been identified as potential areas for harvesting wind energy. Figure 3 shows Q-Q plots of the Weibull distributions of the data from the two provinces. Furthermore, the Akaike Information Criterion (AIC) and

**Table 2  Comparison results of the 95% two-sided confidence intervals for the difference between the coefficients of variation of Weibull distributions for $a = 2$.**

| $(n, m)$ | $k$ | Coverage probability (Expected length) | | | | | |
|---|---|---|---|---|---|---|---|
| | | $CI_{\delta(gci)}$ | $CI_{\delta(pb)}$ | $CI_{\delta(bs)}$ | $CI_{\delta(m.HR)}$ | $CI_{\delta(MCMC)}$ | $CI_{\delta(HPD)}$ |
| (10,10) | 0.5 | 0.9512 | 0.6242 | 0.6852 | 0.8266 | 0.9368 | 0.9466 |
| | | (7.9720) | (1.6763) | (1.6970) | (2.3414) | (5.2853) | (4.5489) |
| | 1 | 0.9514 | 0.9630 | 0.9034 | 0.9632 | 0.9338 | 0.9674 |
| | | (2.3568) | (1.1613) | (1.1622) | (1.3561) | (1.8368) | (1.7938) |
| | 2 | 0.9430 | 0.8992 | 0.8744 | 0.9330 | 0.9300 | 0.9482 |
| | | (1.7036) | (0.9226) | (0.9261) | (1.0758) | (1.3570) | (1.2881) |
| | 2.5 | 0.9476 | 0.8730 | 0.8622 | 0.9128 | 0.9348 | 0.9396 |
| | | (1.6354) | (0.8877) | (0.8907) | (1.0342) | (1.3011) | (1.2200) |
| | 4 | 0.9474 | 0.8532 | 0.8490 | 0.8958 | 0.9294 | 0.9290 |
| | | (1.5569) | (0.8388) | (0.8411) | (0.9825) | (1.2356) | (1.1332) |
| | 9 | 0.9472 | 0.8552 | 0.8830 | 0.9058 | 0.9402 | 0.9388 |
| | | (0.6355) | (0.5167) | (0.5229) | (0.5159) | (0.5988) | (0.5770) |
| (10,20) | 0.5 | 0.9500 | 0.7112 | 0.8002 | 0.8624 | 0.9398 | 0.9596 |
| | | (3.6607) | (1.5948) | (1.1618) | (1.7329) | (3.0961) | (2.9601) |
| | 1 | 0.9502 | 0.9366 | 0.9000 | 0.9358 | 0.9394 | 0.9576 |
| | | (1.8261) | (1.0249) | (1.0273) | (1.1529) | (1.4955) | (1.451) |
| | 2 | 0.9512 | 0.8626 | 0.8586 | 0.9088 | 0.9380 | 0.9370 |
| | | (1.5731) | (0.8627) | (0.8655) | (1.0039) | (1.2633) | (1.1696) |
| | 2.5 | 0.9470 | 0.8464 | 0.8390 | 0.8936 | 0.9330 | 0.9260 |
| | | (1.5644) | (0.8484) | (0.8518) | (0.9886) | (1.2476) | (1.1432) |
| | 4 | 0.9604 | 0.8326 | 0.8300 | 0.8782 | 0.9324 | 0.9186 |
| | | (1.5208) | (0.8193) | (0.8219) | (0.9596) | (1.2102) | (1.0956) |
| | 9 | 0.9544 | 0.8356 | 0.8326 | 0.8840 | 0.9382 | 0.9204 |
| | | (1.5123) | (0.8095) | (0.8134) | (0.9496) | (1.1960) | (1.0712) |
| (20,20) | 0.5 | 0.9538 | 0.6378 | 0.7808 | 0.8274 | 0.9440 | 0.9484 |
| | | (3.2994) | (1.5021) | (1.5258) | (1.5847) | (2.8817) | (2.6754) |
| | 1 | 0.9478 | 0.9510 | 0.9156 | 0.9420 | 0.9402 | 0.9598 |
| | | (1.2330) | (0.8735) | (0.8781) | (0.9134) | (1.1231) | (1.1132) |
| | 2 | 0.9514 | 0.9008 | 0.9038 | 0.9346 | 0.9420 | 0.9526 |
| | | (0.9275) | (0.6855) | (0.6899) | (0.7258) | (0.8490) | (0.8271) |
| | 2.5 | 0.9512 | 0.8846 | 0.8936 | 0.9220 | 0.9414 | 0.9460 |
| | | (0.8922) | (0.6606) | (0.6657) | (0.6971) | (0.8163) | (0.7900) |
| | 4 | 0.9506 | 0.8738 | 0.8844 | 0.9186 | 0.9430 | 0.9468 |
| | | (0.8582) | (0.6286) | (0.6334) | (0.6660) | (0.7847) | (0.7507) |
| | 9 | 0.9486 | 0.8564 | 0.8692 | 0.9058 | 0.9370 | 0.9314 |
| | | (0.8336) | (0.6079) | (0.6133) | (0.6444) | (0.7584) | (0.7186) |
| (30,30) | 0.5 | 0.9540 | 0.6384 | 0.8140 | 0.8240 | 0.9494 | 0.9486 |
| | | (2.3309) | (1.3775) | (1.4077) | (1.2727) | (2.1542) | (2.0485) |
| | 1 | 0.9482 | 0.9486 | 0.9278 | 0.9346 | 0.9458 | 0.9600 |
| | | (0.9231) | (0.7463) | (0.7532) | (0.7347) | (0.8719) | (0.8667) |

**Table 2** (*continued*)

| (n, m) | k | Coverage probability (Expected length) | | | | | |
|---|---|---|---|---|---|---|---|
| | | $CI_{\delta(gci)}$ | $CI_{\delta(pb)}$ | $CI_{\delta(bs)}$ | $CI_{\delta(m.HR)}$ | $CI_{\delta(MCMC)}$ | $CI_{\delta(HPD)}$ |
| | 2 | 0.9570 | 0.9056 | 0.9134 | 0.9316 | 0.9508 | 0.9520 |
| | | (0.7005) | (0.5752) | (0.5798) | (0.5827) | (0.6629) | (0.6512) |
| | 2.5 | 0.9568 | 0.8922 | 0.9074 | 0.9314 | 0.9530 | 0.9536 |
| | | (0.6806) | (0.5584) | (0.5636) | (0.5635) | (0.6437) | (0.6293) |
| | 4 | 0.9484 | 0.8700 | 0.8902 | 0.9090 | 0.9404 | 0.9376 |
| | | (0.6537) | (0.5371) | (0.5435) | (0.5374) | (0.6176) | (0.5992) |
| | 9 | 0.9528 | 0.8602 | 0.8876 | 0.9068 | 0.9436 | 0.9378 |
| | | (0.6341) | (0.5199) | (0.5262) | (0.5194) | (0.5916) | (0.5747) |
| (30,50) | 0.5 | 0.9532 | 0.6688 | 0.8768 | 0.8408 | 0.9496 | 0.9568 |
| | | (1.7003) | (1.2713) | (1.3092) | (1.0311) | (1.6218) | (1.5842) |
| | 1 | 0.9540 | 0.9416 | 0.9306 | 0.9352 | 0.9500 | 0.9576 |
| | | (0.8051) | (0.6830) | (0.6899) | (0.6549) | (0.7671) | (0.7609) |
| | 2 | 0.9504 | 0.8884 | 0.9036 | 0.9200 | 0.9464 | 0.9444 |
| | | (0.6717) | (0.5544) | (0.5595) | (0.5569) | (0.6366) | (0.6212) |
| | 2.5 | 0.9512 | 0.8738 | 0.8988 | 0.9168 | 0.9462 | 0.9412 |
| | | (0.6576) | (0.5405) | (0.5464) | (0.5436) | (0.6227) | (0.6055) |
| | 4 | 0.9528 | 0.8776 | 0.8914 | 0.9124 | 0.9474 | 0.9436 |
| | | (0.6435) | (0.5277) | (0.5341) | (0.5288) | (0.6088) | (0.5888) |
| | 9 | 0.9472 | 0.8552 | 0.8830 | 0.9058 | 0.9402 | 0.9388 |
| | | (0.6355) | (0.5167) | (0.5229) | (0.5159) | (0.5988) | (0.5770) |
| (50,50) | 0.5 | 0.9532 | 0.6506 | 0.8538 | 0.8144 | 0.9484 | 0.9532 |
| | | (1.6366) | (1.2485) | (0.9732) | (0.9766) | (1.5637) | (1.5147) |
| | 1 | 0.9498 | 0.9378 | 0.9380 | 0.9338 | 0.9470 | 0.9564 |
| | | (0.6698) | (0.6120) | (0.6178) | (0.5627) | (0.6484) | (0.6458) |
| | 2 | 0.9512 | 0.9012 | 0.9158 | 0.9288 | 0.9476 | 0.9486 |
| | | (0.5152) | (0.4690) | (0.4729) | (0.4476) | (0.4990) | (0.4931) |
| | 2.5 | 0.9498 | 0.8882 | 0.9198 | 0.9300 | 0.9478 | 0.9500 |
| | | (0.4994) | (0.4542) | (0.4588) | (0.4320) | (0.4834) | (0.4764) |
| | 4 | 0.9484 | 0.8852 | 0.9108 | 0.9212 | 0.9464 | 0.9436 |
| | | (0.4791) | (0.4367) | (0.4416) | (0.4116) | (0.4637) | (0.4549) |
| | 9 | 0.9496 | 0.8812 | 0.9072 | 0.9142 | 0.9446 | 0.9430 |
| | | (0.4689) | (0.4254) | (0.4299) | (0.4005) | (0.4520) | (0.4416) |
| (50,100) | 0.5 | 0.9546 | 0.7092 | 0.9182 | 0.8404 | 0.9522 | 0.9566 |
| | | (1.1315) | (1.1025) | (1.1526) | (0.7399) | (1.1044) | (1.0913) |
| | 1 | 0.9506 | 0.9314 | 0.9384 | 0.9272 | 0.9508 | 0.9562 |
| | | (0.5698) | (0.5402) | (0.5444) | (0.4860) | (0.5548) | (0.5511) |
| | 2 | 0.9450 | 0.8936 | 0.9102 | 0.9134 | 0.9414 | 0.9390 |
| | | (0.4902) | (0.4502) | (0.4552) | (0.4232) | (0.4749) | (0.4670) |
| | 2.5 | 0.9530 | 0.8856 | 0.9162 | 0.9254 | 0.9524 | 0.9490 |
| | | (0.4802) | (0.4380) | (0.4432) | (0.4141) | (0.4659) | (0.4572) |
| | 4 | 0.9512 | 0.8882 | 0.9138 | 0.9198 | 0.9470 | 0.9454 |
| | | (0.4730) | (0.4317) | (0.4369) | (0.4052) | (0.4579) | (0.4482) |
| | 9 | 0.9504 | 0.8750 | 0.9034 | 0.9094 | – | – |
**Table 2** (*continued*)

| (*n, m*) | *k* | Coverage probability (Expected length) | | | | | |
|----------|-----|------------------|------------------|------------------|--------------------|----------------------|---------------------|
| | | $CI_{\delta(gci)}$ | $CI_{\delta(pb)}$ | $CI_{\delta(bs)}$ | $CI_{\delta(m.HR)}$ | $CI_{\delta(MCMC)}$ | $CI_{\delta(HPD)}$ |
| | | (0.4658) | (0.4256) | (0.4319) | (0.3979) | – | – |
| (100,100) | 0.5 | 0.9468 | 0.6818 | 0.9048 | 0.8158 | 0.9496 | 0.9468 |
| | | (1.0733) | (1.0585) | (1.1081) | (0.6835) | (1.0506) | (1.0325) |
| | 1 | 0.9498 | 0.9374 | 0.9524 | 0.9300 | 0.9496 | 0.9512 |
| | | (0.4518) | (0.4644) | (0.4673) | (0.3944) | (0.4446) | (0.4432) |
| | 2 | 0.9552 | 0.9142 | 0.9438 | 0.9364 | 0.9538 | 0.9544 |
| | | (0.3505) | (0.3527) | (0.3548) | (0.3145) | (0.3449) | (0.3424) |
| | 2.5 | 0.9530 | 0.9043 | 0.9424 | 0.9294 | 0.9504 | 0.9504 |
| | | (0.3393) | (0.3447) | (0.3475) | (0.3032) | (0.3340) | (0.3310) |
| | 4 | 0.9472 | 0.8916 | 0.9320 | 0.9166 | 0.9446 | 0.9428 |
| | | (0.3259) | (0.3323) | (0.3351) | (0.2889) | (0.3206) | (0.3170) |
| | 9 | 0.9484 | 0.8944 | 0.9304 | 0.9170 | – | – |
| | | (0.3189) | (0.3253) | (0.3286) | (0.2810) | – | |

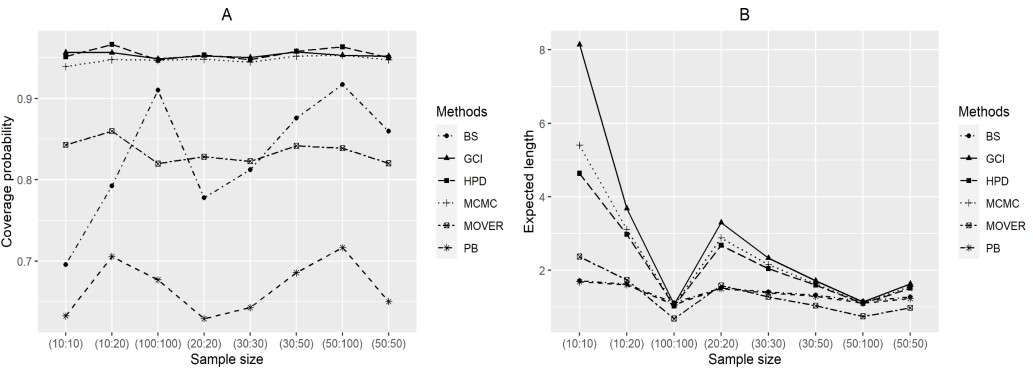

**Figure 2 Graphs to compare the performance of the methods with varying sample size in terms of (A) coverage probability and (B) expected length.**

Bayesian Information Criterion (BIC) values reported in Table 3 indicate that these two datasets fit Weibull distributions because the latter have the smallest values for both criteria. The statistical summary of the estimates of the parameters for these datasets is $(n, m) = (12, 11)$, $\hat{a}_X = 1.7621$, $\hat{a}_Y = 4.5839$, $\hat{k}_X = 1.4832$, $\hat{k}_Y = 9.0794$, $\hat{\lambda}_X = 0.6860$ and $\hat{\lambda}_Y = 0.1317$, while the actual difference between the coefficients of variation of the two Weibull distributions $\hat{\delta} = 0.5543$. The confidence intervals based on the proposed methods are given in Table 4. The results show that the Bayesian-HPD interval performed well in terms of the coverage probability and expected length when the sample sizes are small, which supports the simulation results. Finally, Fig. 4 shows a trace plot of the generated $\delta$ value.

## Example 2

Data on wind speeds measured at 90-meter wind energy potential stations in the southern and northeastern regions of Thailand were collected in April–May, 2019

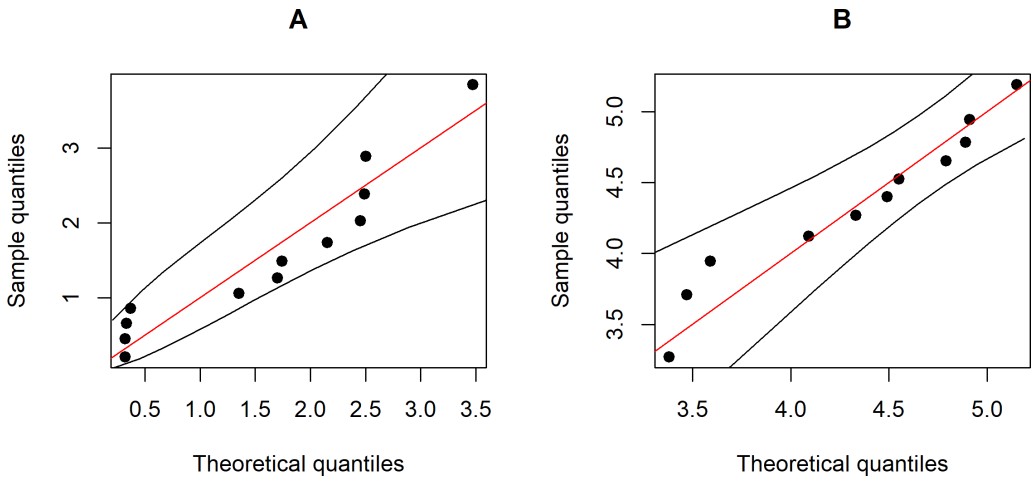

**Figure 3** The Weibull Q-Q plot for the wind speed data of two provinces, Thailand: (A) Trad Province (B) Chonburi Province.

**Table 3** The AIC and BIC values of the wind speed data of Trad province and Chonburi province.

| Method | Trad | | Chonburi | |
|---|---|---|---|---|
| | AIC | BIC | AIC | BIC |
| Weibull | 37.0272 | 37.9970 | 22.8054 | 23.6011 |
| Exponential | 37.2675 | 37.7524 | 56.2471 | 56.6450 |
| Gamma | 37.5735 | 38.5433 | 24.0780 | 24.8737 |
| Log-normal | 39.1594 | 40.1292 | 24.3327 | 25.1285 |

**Table 4** The 95% confidence intervals for the difference between the coefficients of variation of the wind speed data of Trad and Chonburi provinces, Thailand.

| Method | Confidence intervals for $\delta$ | | |
|---|---|---|---|
| | Lower | Upper | Length |
| GCI | 0.3316 | 1.1013 | 0.7697 |
| MOVER | 0.2398 | 0.8689 | 0.6291 |
| PB | 0.2081 | 0.8036 | 0.5155 |
| BS | 0.2531 | 0.8555 | 0.6024 |
| MCMC | 0.2912 | 1.0095 | 0.7183 |
| HPD | 0.2600 | 0.9216 | 0.6616 |

by the Department of Alternative Energy Development and Efficiency, Ministry of Energy (https://www.dede.go.th/more_news.php?cid=501). The summary statistics for the southern region are $n = 12$, $\hat{a}_X = 2.5334$, $\hat{k}_X = 2.3956$ and $\hat{\lambda}_X = 0.4441$ and the northeastern region are $m = 20$, $\hat{a}_Y = 4.4608$, $\hat{k}_Y = 3.8457$ and $\hat{\lambda}_Y = 0.2906$. The actual difference between the coefficients of variation of the Weibull distributions of these datasets $\hat{\delta} = 0.1538$. Weibull Q-Q plots of these data are shown in Fig. 5 and the correctness of fitting the data to Weibull distributions in terms of the smallest AIC and BIC values are
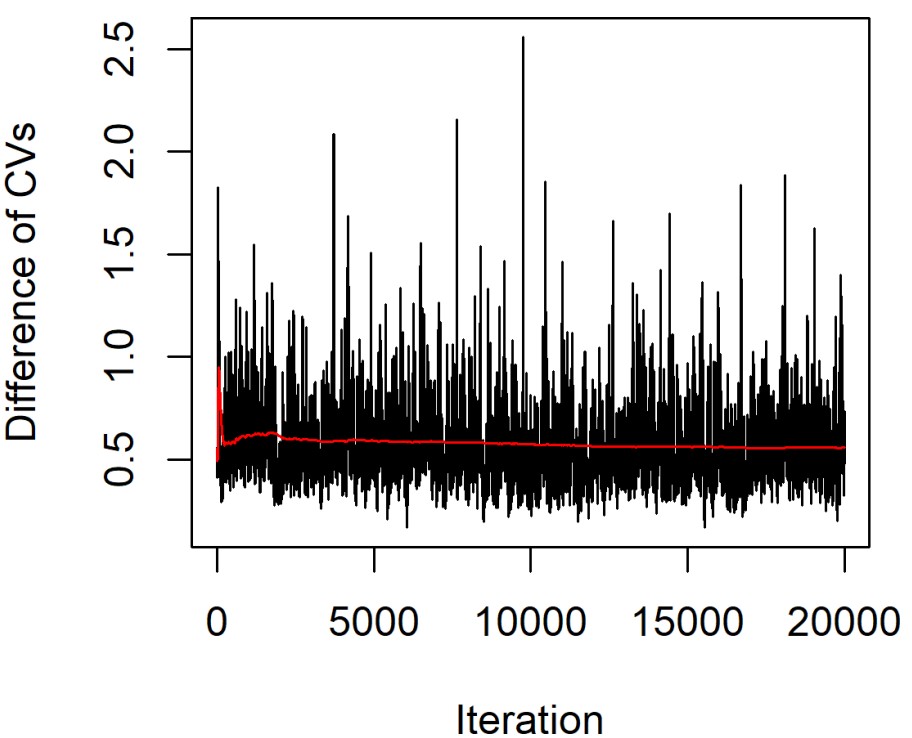

**Figure 4** Plot of generated δ of example 1 vs. iteration of the MCMC algorithm.

**Table 5** The AIC and BIC values of the wind speed data of the southern and northeastern regions of Thailand.

| Method | Southern | | Northeastern | |
|---|---|---|---|---|
| | AIC | BIC | AIC | BIC |
| Weibull | 37.9176 | 38.8871 | 68.3758 | 70.3673 |
| Exponential | 45.4623 | 45.9472 | 97.6711 | 98.6669 |
| Gamma | 39.2535 | 75.5790 | 71.5482 | 73.5397 |
| Log-normal | 41.5439 | 42.5137 | 73.5875 | 40.2233 |

reported in Table 5. The 95% confidence intervals for δ by applying the six methods (Table 6) indicate that the Bayesian-HPD interval performed well in terms of the coverage probability and expected length when the difference between the coefficients of variation was small, which is once again consistent with the simulation study results. Finally, the trace plot of the generated δ value are presented in Fig. 6.

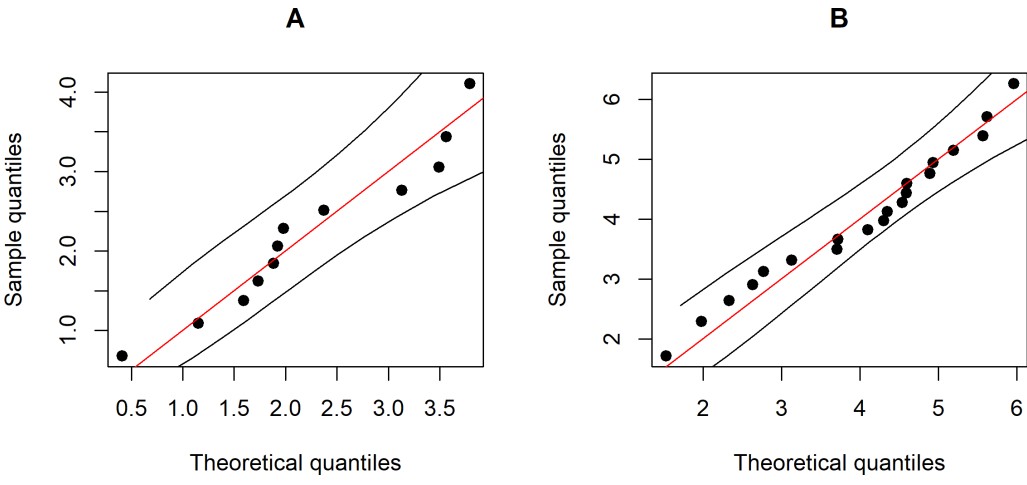

**Figure 5** The Weibull Q-Q plot for the wind speed data of two Thailand's regions: (A) Southern (B) Northeastern.

**Table 6** The 95% confidence intervals for the difference between the coefficients of variation of the wind speed data of the southern and northeastern regions of Thailand.

| Method | Confidence intervals for $\delta$ | | |
|---|---|---|---|
| | Lower | Upper | Length |
| GCI | −0.0270 | 0.4807 | 0.5077 |
| MOVER | −0.0678 | 0.3755 | 0.4435 |
| PB | −0.0450 | 0.3763 | 0.4213 |
| BS | −0.0553 | 0.3630 | 0.4183 |
| MCMC | −0.0241 | 0.4727 | 0.4768 |
| HPD | −0.0445 | 0.4267 | 0.4712 |

# DISCUSSION

It can be seen from the results of the study that GCI performed well in almost all cases since its coverage probability was greater than or close to the nominal confidence level. For the Bayesian methods, Bayesian-MCMC and Bayesian-HPD interval produced similar results and performed well when $k_Y$ (the difference between the coefficients of variation) was small, zero, or negative. For large sample sizes, the Bayesian methods were not suitable when $k_Y$ was large. Moreover, the expected length of the Bayesian-HPD interval was always shorter than Bayesian-MCMC. In relation to the expected lengths, they tended to decrease when the sample sizes and/or $k_Y$ increased.

# CONCLUSIONS

This paper proposed six methods for deriving the confidence interval for the difference between the coefficients of variation of Weibull distributions: the GCI method, the Bayesian-MCMC method, the Bayesian-HPD interval, MOVER based on Hendricks and Robey's confidence interval, the percentile bootstrap method, and the bootstrap confidence

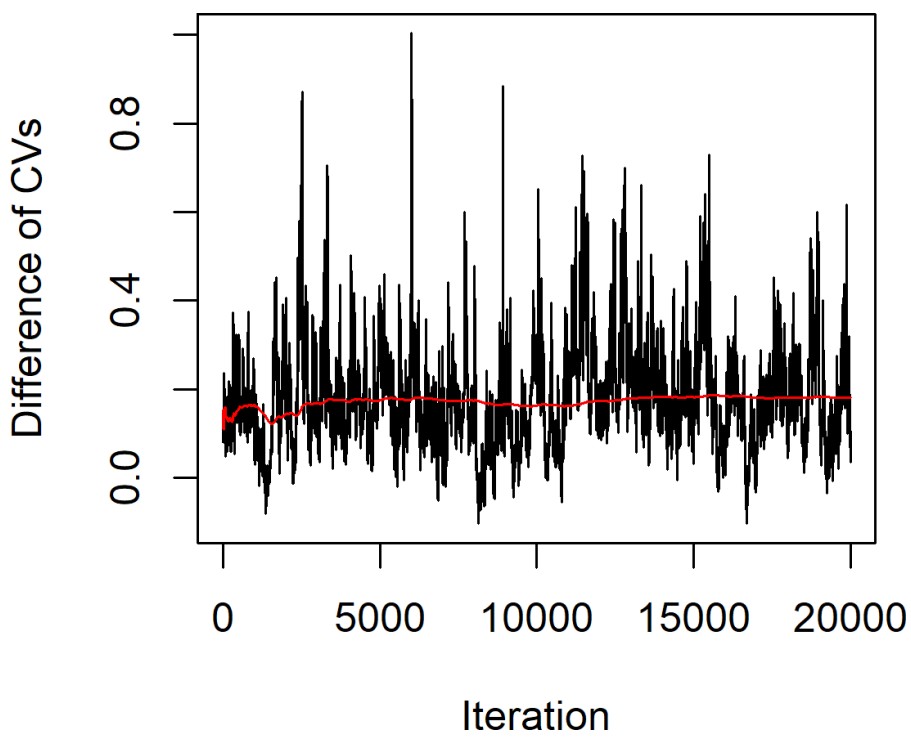

**Figure 6** Plot of generated $\delta$ of example 2 vs. iteration of the MCMC algorithm.

interval with standard errors. The performances of these methods were compared using the coverage probability and the expected length. The results of simulation studies showed that the GCI method and the Bayesian-HPD method were the best in different scenarios. The Bayesian-HPD was preferable when shape parameter $k_Y$ was small. The percentile bootstrap method can be used when the sample sizes are small and $k_Y = 1$. However, MOVER based on Hendricks and Robey's confidence interval and the bootstrap confidence interval with standard errors are not recommended since these methods yielded coverage probabilities under the nominal confidence level of 0.95 for almost all cases.

### Funding
This research was funded by King Mongkut's University of Technology North Bangkok. Contract no. KMUTNB-PHD-62-04 The funders had no role in study design, data collection and analysis, decision to publish, or preparation of the manuscript.

### Grant Disclosures
The following grant information was disclosed by the authors:
King Mongkut's University of Technology North Bangkok: KMUTNB-PHD-62-04.

### Competing Interests
The authors declare there are no competing interests.

## Author Contributions

- Manussaya La-ongkaew conceived and designed the experiments, performed the experiments, analyzed the data, prepared figures and/or tables, authored or reviewed drafts of the paper, and approved the final draft.
- Sa-Aat Niwitpong conceived and designed the experiments, analyzed the data, authored or reviewed drafts of the paper, and approved the final draft.
- Suparat Niwitpong conceived and designed the experiments, performed the experiments, analyzed the data, authored or reviewed drafts of the paper, and approved the final draft.

## Data Availability

The raw data showing wind speeds and the R code are available in the Supplemental Files.

## Supplemental Information

Supplemental information for this article can be found online at http://dx.doi.org/10.7717/peerj.11676#supplemental-information.

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
