# Peer review of "Confidence intervals for the difference between the coefficients of variation of Weibull distributions for analyzing wind speed dispersion"

_PeerJ, doi:10.7717/peerj.11676_

## Round 0.1 · original submission · Major Revisions

Kindly attend to each of the points the reviewers' comments have raised. Also, please seek the help of a professional editor to correct the grammar mistakes in the document.

Reviewer 1 ·

Basic reporting

There are numerous spelling and grammar issues that distract the reader. To maximize readership among the international audience, the manuscript should be proofread to fix those issues.

Experimental design

no comment

Validity of the findings

no comment

Additional comments

The study presented new confidence intervals for the difference between the coefficients of variations of Weibull distributions using different methods. While generally successful, some modifications are needed.

General comment:
There are numerous spelling and grammar issues that distract the reader. To maximize readership among the international audience, the manuscript should be proofread to fix those issues.

Specific Comments:

1. The first paragraph of the introduction is generally good, but including citations to some stated information would add credibility. For example, in line 30, the authors say that wind energy usage is limited in Thailand; citing a governmental report or another paper would help. Also, a minor comment in this paragraph in line 26 where the authors say that solar, wind, and hydro would be “developed”: the word “developed” is not the right word as these sources are already there but need to be efficiently harvested or utilized. So a more accurate word needs to be used.

2. In the first line of the “Methods” section, specifying that X follows the “2-Parameter Weibull” function is more accurate as this function can have 1 or 3 parameters.

3. In the second paragraph of the Methods section, I assume MLE refers to the maximum likelihood estimation. The abbreviation needs to be clearly defined.

In line 82, the purpose of Algorithm 1, which is used to find k hat and a hat, needs to be properly stated to ensure a smooth flow as the algorithm appears to be disconnected from the previous paragraph.

4. In line 84, how is Z bar computed if S and ku are unknowns? Also what does subscript “u” mean in ku? A clarification is needed

5. In line 96: “Definition 1” gives the impression that a second definition is going to be stated later in the manuscript which is not the case. It should be removed.

6. In line 103, a definition of alpha and the possible range are required.

7. In the line after 109, what does GPQ stand for?

8. Line 138 and 151: The steps are not clearly stated.

9. Line 141 and 154: why are paranthesis used?

10. Line 252 and 277: A citation of the governmental reports would give more credibility to the authors’ results.

11. Figure 4: I assume the x-axis title is iteration?

Reviewer 2 ·

Basic reporting

The authors presented confidence intervals for the difference between the coefficients of variation of Weibull distributions using six different statistical methods. The performance of each method was assessed based on analyzing the coverage probabilities and the expected lengths. Real wind speed data were used as an empirical application for the proposed confidence intervals.

Based on my evaluation, the work was properly presented and the conclusions were clearly stated. The work is appreciable given the fact that Weibull distributions are widely accepted in many research areas and that the coefficient of variation (which is the parameter of interest in this paper) is used for measuring variation in data.

Experimental design

The experimental design is well structured. The statistical algorithms were clearly described for each of the six methods. Some minor changes, additions, modifications, are however needed. These are stated in the Comments for the Author section.

Validity of the findings

In general, the results were properly presented in terms of figures and tables (although some minor changes are needed). Among the different used statistical methods, a clear comparison was made yielding major conclusions.

Additional comments

Note that some lines in the text are not numbered and therefore some of my comments for such ones will refer instead to the closest numbered line or equation.

Literature Review Comments:
1) Why the earlier work published by the second and the third author, which is related to the confidence intervals for single coefficient of variation of Weibull distribution, is not mentioned in your current literature review?
2) A brief explanation for why considering the difference in the coefficient of variation over the single coefficient of variation in this work is needed.
3) You need to emphasize before line 75 that no previous work has considered providing the confidence intervals for the difference in the coefficient of variation of Weibull distribution. This will make the novelty of your work clear to the reader.



Algorithm Comments:
4) Place a sentence before each algorithm stating the purpose of the algorithm. For example: “The following algorithm is used to compute…”
5) End all the steps of your algorithms with a period.
6) Number your steps as 1. 2. 3. … instead of step1, step2, step3, …
7) Line 141: remove the parenthesis.

Figures and Tables Comments:
8) Add a row before the CI’s row in Table 1 and name it “Coverage probability (Expected length)”. Then delete the bottom caption of this table.
9) You might need to try increasing the font size of the x-label, y-label, and legend in figures 1 and 2. They need to be easily readable. The rest of the figures are clear.
10) The caption of figure 1 should be indeed the one for figure 2 and vice versa.
11) End all figure captions and table captions with period.
12) Change “intenration” to “iteration” in the caption of figure 4.


Syntax and Grammar Comments:
13) Under equation 23: change “it is assume” to “it is assumed”.
14) Line 29: change “its use reduces” to “it reduces”.
15) Lines 32-33: change “and the strength of the wind varies throughout the year” to “and wind strength variation throughout the year”.
16) Line 93: add “was” before “introduced”.
17) Page 3: remove “respectively” in the following sentence “Similarly, let λY be the coefficient of variation of Y, respectively.”
18) Line 104: change “be” to “is”.
19) Line 107: change “be” to “are”.
20) Line 147: change “be” to “is”.
21) Line 167: change “form” to “from”.
22) Line 170: change “be” to “is”.
23) End the sentences in lines 217, 219, 220, and 221 with a period.
24) Line 264: change “Figure 3 exhibits” to “Figure 3 shows”.

Other Comments:
25) Lines 66-67: place all the citations as such: (Vangel, 1996; Tian, 2005; Mahmoud and and Hassani, 2009; Donner and Zou, 2010).
26) Define “MLEs” as maximum likelihood estimators when you first introduce it in the text in the last line of page 2. Change your sentence accordingly: “The maximum likelihood estimators (MLEs) can be obtained from Cohen (1965)…”.
27) Similarly define “GPQs” as generalized pivotal quantities after line 109.

---

## Round 0.2 · accepted · Accept

The reviewers have now accepted the modifications introduced to the revised manuscript and as such the manuscript is accepted for publication.

Reviewer 1 ·

Basic reporting

no comment

Experimental design

no comment

Validity of the findings

no comment

Additional comments

The authors have addressed the comments outlined in the first review.